# Phylogenetic Analysis and Serological Investigation of Porcine Circovirus Indicates Frequent Infection with Various Subtypes

**DOI:** 10.3390/ijms242115850

**Published:** 2023-11-01

**Authors:** Qianling Peng, Jiqiang Shi, Yifei Lang, Yulan Zhu, Xiaobo Huang, Sanjie Cao, Qigui Yan, Shan Zhao

**Affiliations:** 1College of Veterinary Medicine, Sichuan Agricultural University, Chengdu 611130, China; 2Key Laboratory of Animal Disease and Human Health of Sichuan Province, Sichuan Agricultural University, Chengdu 611130, China

**Keywords:** PCV, phylogeny, serology, Cap, prevalence

## Abstract

Porcine circoviruses (PCVs) are notorious for triggering severe diseases in pigs and causing serious economic losses to the swine industry. In the present study, we undertook a comprehensive approach for the investigation of PCV prevalence, including the phylogenetic analysis of obtained PCV sequences, the determination of major circulating genotypes and serological screening based on different recombinant Cap proteins with specific immunoreactivity. Epidemiological surveillance data indicate that PCV2d and PCV3a are widely distributed in Southwest China, while PCV4 has only sporadic circulation. Meanwhile, serological investigations showed high PCV2 antibody positivity in collected serum samples (>50%), followed by PCV4 (nearly 50%) and PCV3 (30–35%). The analysis supports different circulation patterns of PCV2, PCV3 and PCV4 and illustrates the PCV2/PCV3 genetic evolution characteristics on a nationwide basis. Taken together, our findings add up to the current understanding of PCV epidemiology and provide new tools and insight for PCV antiviral intervention.

## 1. Introduction

Porcine circovirus (PCV) is a group of circular single-stranded DNA (ssDNA) viruses that belongs to the genus *Circovirus* of the *Circoviridae* family, which represents the smallest viruses able to infect the swine population [1,2]. Thus far, PCV is divided into four known species, namely PCV1, PCV2, PCV3 and PCV4 [3]. PCV1 was first recognized as a continuous contaminant of porcine kidney cell line in 1974 [4]. It was primarily a small, spherical virus particle and deemed non-pathogenic to pigs [5]. Contrary to PCV1, PCV2 is a circulating pathogen that causes porcine circovirus-associated disease (PCVAD), including postweaning multisystemic wasting syndrome (PMWS), porcine dermatitis and nephropathy syndrome (PDNS), reproductive disorders and respiratory disease [6,7,8,9]. In 2015, a novel porcine circovirus was identified in mummified fetuses and sows and then categorized as a new circovirus species, namely PCV3 [10]. A few years later, another novel circovirus was identified through sequencing and molecular characterization, which was designated as PCV4 [11]. PCV3 and PCV4 infections are generally associated with clinical symptoms such as PDNS, together with respiratory and enteric signs. The rapid emergence and maintained circulation of PCV3 and PCV4 warrant future research into circovirus etiology and the development of antiviral interventions, such as vaccines and virus-targeting drugs.

The high nucleotide substitution rate and the recombination events that occur in PCV2 are the main driving forces behind the development of new viral genotypes [12]. For uniform classification, PCV2 can be classified to eight genotypes: PCV2a-PCV2h [1]. In contrast to PCV2, PCV3 and PCV4 classifications are poorly supported due to the limited number of sequences and the corresponding biological data, and each species can only be roughly divided into two genotypes, namely PCV3a and PCV3b and PCV4a and PCV4b [13,14]. The current circulating PCVs have a DNA genome of 1769–1776 nucleotides (nt) (PCV2), 1999–2001 nt (PCV3) or 1770 nt (PCV4), which encompasses two ORFs, namely ORF1 and ORF2 [1]. ORF1 is located on the sense strand of the genome, encoding the viral replication-related (Rep) protein, which is essential for viral DNA replication [15,16]. ORF2 is located on the anti-sense strand of the genome and encodes the capsid (Cap) protein [17]. As the main structural protein, Cap protein contains highly conserved linear and conformational epitopes, induces host immunity and mediates immune evasion during PCV infection [18,19,20,21]. Therefore, Cap protein serves as a versatile tool for molecular diagnosis and serological surveillance.

In the past few years, many studies have shed light on PCV molecular epidemiology and evolutionary dynamics, showing the high prevalence of PCV2 and PCV3 and the limited prevalence of PCV4 in China [17,22,23,24,25]. However, PCV seroprevalence is much less investigated. Recombinant Cap protein had been exploited as a target antigen by researchers for the detection of specific PCV antibodies, but the seroprevalence of different PCV genotypes has hardly been thoroughly clarified in contemporary surveys [26,27,28].

In this study, we conducted a detailed prevalence investigation of PCV2, PCV3 and PCV4 from both phylogenetic and serological perspectives. Based on clinical samples from partial regions of China, our research illustrated the genetic diversity of PCV2 and PCV3 and elucidated the seroprevalence of different PCV genotypes. The results obtained shall benefit the current understanding of PCV epidemiology and pave the way for the future development of novel vaccines and antiviral interventions.

## 2. Results

### 2.1. Lasting Widespread Distribution of PCV2 and PCV3 and Sporadic Prevalence of PCV4 Revealed in the Southwest of China

Among the 658 porcine clinical samples included in the present study, which were mainly collected from 12 cities in Sichuan province between 2019 and 2022 (Figure 1A), the positive rate of PCV2, PCV3 and PCV4 were 28.27% (186/658), 30.7% (202/658) and 1.98% (13/658), respectively. Co-infection with PCV2, PCV3 and PCV4 was identified in 1.67% (11/658) of the samples tested (Figure 1B).

When analyzed based on sample type, the positive rates of PCV2, PCV3 and PCV4 were 69.78% (97/139), 51.08% (71/139) and 2.16% (3/139) in tissue samples, respectively, which were higher for each corresponding virus compared to those observed for serum samples (detailed in Table 1). The yearly positive rates of PCV2 and PCV3 were stable during 2019–2022, while PCV4 positivity was detected only in samples collected in 2022 (Table 2).

### 2.2. PCV2d and PCV3a as the Predominant Circulating Genotypes in the Southwest of China

For ORF2 amplification, samples with Ct values lower than 30 were selected. In total, 60 PCV2 ORF2 genes and 48 PCV3 ORF2 genes were successfully amplified, sequenced and then subjected to further phylogenetic and phylogeographic analysis.

As shown by the ML trees of PCV2 (Figure 2A–D), six PCV2 genotypes can be identified in China, namely PCV2a, PCV2b, PCV2d, PCV2e, PCV2f and PCV2h, which is in accordance with previous reports [22,29,30]. In detail, the ML tree revealed that PCV2a, PCV2b and PCV2d were circulating throughout China in the past few decades. In the present study, 4 strains of PCV2 were clustered in PCV2a, 17 strains were clustered in PCV2b and 39 strains were clustered in PCV2d. The results indicate that PCV2a, PCV2b and PCV2d are widely distributed in Southwest China, while the main genotype is PCV2d. Meanwhile, 48 PCV3 ORF2 sequences were analyzed together with reference sequences. PCV3 comprises two major genotypes, namely PCV3a and PCV3b, and several intermediate strains (IM), as reported in [14]. An ML tree based on PCV3 ORF2 shows that 35 strains from Southwest China in this study belong to PCV3a, while 13 strains belong to PCV3b. PCV3a is the most widespread genotype in Southwest China (Figure 2E). To explore which essential residues are changed in different PCV2 and PCV3 genotypes, we performed amino acid sequence alignments of the amino acid residues of each type (Figure 3A,B).

### 2.3. High Seroprevalence of PCV2 as the Result of Both Vaccination and Natural Infection

Concentrations of recombinant proteins were assessed via Nanodrop (ThermoFisher Scientific Inc., Waltham, MA, USA). Prior to serological studies, the recombinant Cap proteins expressed in this study was first identified using SDS-PAGE and Western blot analysis. The results indicated that the Cap proteins were of the expected molecular weight (Figure 4A) and exhibited reactivity with both the anti-Strep tag antibody and different PCV-positive sera (Figure 4B). Next, we screened a collection of porcine serum samples (*n* = 165) from 2019 (*n* = 55) and 2021 (*n* = 110) via indirect ELISA against six different PCV cap antigens in order to assess the seroprevalence of PCVs in Southwest China (Figure 5A). In total, the majority of antibody responses were against PCV-2a (103/165, 62.42%), followed by PCV-2b (95/165, 57.58%), PCV-2d (94/165, 56.97%), PCV4 (81/165, 49.09%), PCV-3b (57/165, 34.55%) and PCV-3a (51/165, 30.91%) (Figure 5B,C). In order to elucidate the trends of antibody levels on time scales, we also compared the positive rates of samples extracted in 2019 and 2021 (Figure 5D).

## 3. Discussion

At present, PCV2 and PCV3 are causing significant economic loss to the swine industry, as they are able to infect both healthy and diseased pigs through horizontal or vertical transmission [31,32,33,34]. In 2019, PCV4 was discovered in China and subsequently spread to several regions, though the current knowledge regarding its distribution and pathogenicity is still limited. Therefore, due to the high prevalence rate of PCV2/PCV3 and the potential impact of PCV4, it is essential to characterize the prevalence and evolution features of these pathogens. To this end, in this study, we investigated the evolution trend and seroprevalence of different PCV genotypes using the phylogenetic approach and a newly established serological methodology based on the recombinant Cap protein of different genotypes.

In this study, 658 samples from four provinces were collected during 2019–2022, most of which were from Southwest China. Samples were detected via real-time PCR, and the prevalent rates of PCV2, PCV3 and PCV4 were 28.27% (186/658), 30.7% (202/658) and 1.98% (13/658), respectively. The epidemiology of PCV2 and PCV3 is similar to previous reports, while both our data and those of the study by Xu et al. indicated a relatively limited spread of PCV4 in Southwest China [35]. Sixty-four samples showed co-infection between different PCV types, among which 53 samples were co-infected with PCV2 and PCV3, and 11 samples were coinfected with PCV2, PCV3 and PCV4. Both PCV2 and PCV3 infection can inhibit immune responses, contributing to immune evasion and enhancing immunosuppression [36,37] Therefore, PCV2 and PCV3 coinfection may lead to the stronger immunosuppression of the host, causing more severe clinical symptoms, while rendering pigs vulnerable to other pathogens. To further study the evolution and phylogeographic features of PCV2 and PCV3 on a countrywide basis, we collected all available PCV2/PCV3 sequences through rigorous data selection. Among 60 PCV2 strains from southwest China, only three genotypes were identified, namely PCV2a, PCV2d and PCV2b, while the proportions of each were 6.67% (4/60), 28.33% (17/60) and 68% (39/60). This result suggested that PCV2d was predominant in Southwest China. According to ML trees based on PCV2 ORF2 gene (Figure 2A–D), PCV2b was the main genotype in China before 2009, which is consistent with a previous report (Figure 2A) [24]. Afterwards, the prevalent genotype shifted from 2b to 2d around 2009, and 2d remained has prevalent until now (Figure 2B–D). In China, PCV2 vaccines were acquired from Germany in 2009, the same year when the genotype changed from PCV2b to 2d, which matches a previous analysis stating that vaccination strategies would impair viral evolutionary patterns [38]. In addition to PCV2a, PCV2b and PCV2d, PCV2f, PCV2h and PCV2e were identified in China, but the prevalence rate was not high. A previous report defined 2f and 2h as IM groups, namely IM3 and IM1, considering them as relic and inactivated strains [39]. Therefore, there were very few reports of 2f and 2h in China. PCV2e strains showed high heterogeneity with other PCV2 genotypes in the ORF2 gene. Although PCV2e strains were found in three provinces in 2017 and 2022, they did not spread to other regions, which is probably due to the fact that they were much less infectious than other three predominant genotypes [22,40,41]. Unlike the classification of PCV2, a unified classification criteria for PCV3 was proposed recently, while the reference sequences and all 48 PCV3 ORF2 sequences in this study were classified into two stable clades (PCV3a, PCV3b) and one flexible IM group [42]. Although the sequences of six strains from Fujian province were clustered with PCV3b, they cannot accurately represent the overall prevalence of PCV3 in Fujian province due to the limited number of samples. The remaining 42 sequences all belonged to samples from Southwest China, with 7 sequences clustered into PCV3a and 35 sequences clustered into PCV3b.

PCV sequences obtained in the present study mostly clustered with sequences from in East or Central China, such as Shandong, Henan, Hubei and Hebei Provinces. In the last few years, reports investigated the prevalence rates of different areas in China, and it was shown that Henan Province has the highest prevalent rate [13,23]. We estimated that it may due to the fact that Henan imported the most live swine in China and also acted as a live swine trade transportation hub, connecting other regions [43]. Sichuan Province, the biggest live swine trade province in Southwest China, may import live pigs and swine products from other provinces such as Henan, Guangdong and Hubei. Live swine trade transportation has driven the transmission of swine pathogens from one province to another. We inferred that the live swine trade within China has promoted the national distribution of PCV2 and PCV3. A similar situation may also be true for PCV4 as, though its genome was only detected in several provinces, PCV4 might still able to cause nationwide spreading via live swine transportation routes [17].

As the major structural protein of PCV, amino acid sequences of the Cap protein obtained in this study were compared to those of reference strains. Typical motifs 86–91 of PCV2 strains in this study are important for distinguishing between genotypes (PCV2a: ^86^TNKISK^91^; 2b: ^86^SNPRSV^91^; 2d: ^86^S(P)NPL(R)TV^91^). It has been reported that aa 50–81, 113–134, 161–208 and 227–233 in Cap protein were antibody recognition sites, while aa 168–180 was an immune-dominant decoy epitope [44,45]. Figure 3A showed that aa 113–120 of PCV2 ORF2 strains were conserved, while aa 220–225 and 228–233 were highly variable. We hypothesized that this may affect the antigenicity among genotypes of PCV2. In terms of PCV3, aa 24 was pivotal for PCV3 to escape from host immunity and the divergence of PCV3a (^24^A) to PCV3b (^24^V) [14]. Interestingly, EC_FJ202204 belonged to PCV3b with ^24^A instead, which may suggest that EC_FJ202204 was a special strain under the evolution process from 3a to 3b. Meanwhile, A^24^V may not be the only residue to differentiate PCV3a from PCV3b. SW_ZY202102 possessed a unique insertion at aa 208–209, which could be a frameshift mutation due to deletion. Such a deletion also occurred in another unique strain: SW_MS202204.

Cap protein was capable of eliciting neutralizing antibodies in vivo [46]. Therefore, we conducted serological surveys using recombinant Cap proteins as antigens. The expression of complete Cap protein was shown to be difficult, as the nuclear localization signals (NLS) are located at the N-terminus of ORF2 [37,47]. Here, we designed and obtained the NLS-deleted Cap protein of PCV2a, 2b, 2d, PCV3a, 3b and PCV4, and all of them displayed specific reactivity with serum samples. Among the serum samples screened, most positive samples were collected from sows. This scenario was consistent with the infection kinetics of PCV2, as almost all sows were seropositive, with or without clinical signs of PMWS [48]. Among the three genotypes of PCV2, the antibody level of PCV2a was slightly higher than that of 2b and 2d in general (Figure 5C,D). This minor difference is likely the result of the widespread use of PCV2a vaccines [49]. On the other hand, the incomplete protection of PCV2 vaccines also contributed to continuous infection with PCV2. Accordingly, we assumed that viral infection and vaccine application jointly lead to the high antibody positivity of PCV2. Intriguingly, reactivity against PCV4 increased in 2021, which might represent its early PCV4 distribution to other areas since its first identification in 2019, as the antibody level would gradually increase with the transmission of PCV4 in past three years. Co-infection with PCV2, PCV3 and PCV4 was also observed at the antibody level (Figure 5B), confirming the possibility of co-infection. We next analyzed the homologies of the Cap proteins of PCV2, PCV3 and PCV4. The identities between the Cap proteins of PCV3 and PCV4 are less than 25% (Table 3), while the Cap protein of PCV2 showed nearly 45% consistency with PCV4 Cap (Table 3). In the present study, numerous serum samples were confirmed to be only positive for the Cap of one particular genotype (Figure 5A), indicating that the recombinant proteins and serological methods applied in this study are reliable and intra-specific.

## 4. Materials and Methods

### 4.1. Serum and Tissue Samples Collection

From 2019 to 2022, in total, 658 samples were collected across four Provinces of China (Sichuan, Fujian, Zhejiang and Chongqing), including 519 serum samples and 139 tissue samples (spleens, lymph nodes and livers). In total, 658 samples were collected from pigs of different ages, including nursery pigs, fattening pigs, milking sows, breeding boars, gilts and piglets. The majority of the samples were obtained from 12 cities in Sichuan Province. All samples were stored at −80 °C before use. Considering the volume and quality of each serum samples, 165 serum samples were chosen for the investigation of antibodies. PCV2-positive serum samples were identified using a commercial PCV2 AB ELISA kit (MEDIAN, Chuncheon-si, Republic of Korea), while PCV3- and PCV4-positive serum samples were identified using an SYBR Green (Taq Pro Universal SYBR qPCR Master Mix, Vazyme Biotech Co., Ltd., Nanjing, China)-based real-time PCR assay, as described previously [13,50]. Negative serum samples collected from specific-pathogen free pigs were stored in our laboratory, with their quality confirmed via a commercial PCV2 AB ELISA kit (MEDIAN, Chuncheon-si, Republic of Korea) or real-time PCR assay [13,29]. The primers were provided by Sangon Biotech Co., Ltd., Shanghai, China.

### 4.2. Detection of PCVs and Sequencing of ORF2 Gene

Prior to DNA isolation, tissue samples were homogenized in sterile phosphate-buffered saline (PBS), followed by centrifugation at 4000 rpm for 5 min after three freezing–thaw cycles, while the supernatants were collected for DNA extraction. Viral DNA from tissue and serum samples was extracted using Viral DNA Kit (Omega Bio-Tek, Inc., Norcross, GA, USA). The DNA samples were all tested for traces of the PCV2, PCV3 and PCV4 genome through SYBR Green (Taq Pro Universal SYBR qPCR Master Mix, Vazyme Biotech Co., Ltd., Nanjing, China)-based real-time PCR assay, as described previously [13,50]. Samples with Ct values < 35 were considered positive. The real-time PCR were conducted using a C1000 thermal cycler (Bio-Rad Laboratories, Hercules, CA, USA).

Next, PCV2-, PCV3- and PCV4-positive samples were selected for the amplification of the ORF2 gene. Two pairs of primers were designed to amplify the ORF2 genes of PCV2 and PCV3, respectively (Table 4). The PCR reaction was prepared in a total volume of 25 µL: 12.5 µL of 2× GoTaq^®^ Green Master Mix (Promega Biotech Co., Ltd., Madison, WI, USA), 1 µL of each primer, 2 µL of DNA and 8.5 µL of ddH_2_O. The thermal conditions were initiated at pre-denaturation of 95 °C for 5 min, followed by 40 cycles of 95 °C for 5 min, 58 °C (for PCV2 ORF2)/56 °C (for PCV3 ORF2) for 30 s and 72 °C for 30 s, as well as a 5 min extension at 72 °C. The ORF2 gene was amplified using T100 Thermal Cycler (Bio-Rad Laboratories, Hercules, CA, USA). The positive plasmids were standard plasmids of PCV2 and PCV3 preserved in our lab, and positive plasmids for PCV4 were synthesized according to this reference sequence: PCV4 HNU-AHG1-2019 (MK986820); ddH_2_O was set as a negative control at the same time.

Each PCR product was analyzed via Power Pac Universal (Bio-Rad Laboratories, Hercules, CA, USA) using 1.5% agarose gel (Yeasen Biotechnology (Shanghai) Co., Ltd., Shanghai, China). Subsequently, the bands were visualized using the GEL DOC™ XR+ gel documentation system (Bio-Rad Laboratories, Hercules, CA, USA). The 949- and 883-bp PCR products were then purified using the Gel Extraction Kit (Omega Bio-Tek, Inc., Norcross, GA, USA) and sequenced.

### 4.3. Phylogenetic and Phylogeographic Analysis

Considering the difficulty involved in collecting full-length sequences and the similarity between PCV2 ORF2 and whole genome analysis [51]**,** the ORF2 gene was selected for subsequent analysis. Initially, 3906 PCV2 ORF2 and 1311 PCV3 ORF2 domestic sequences were downloaded from GenBank. Reference sequences that lack collection dates or the location of low-quality parts of duplicate copies were removed from the dataset. The final collections of PCV2 and PCV3 ORF2 comprised 2276 and 798 sequences, respectively. Since the prevalent genotype shifted from PCV2b to PCV2d around 2009 [24], and as PCV2e was first identified in China in 2017 [40]**,** we divided all the PCV2 ORF2 sequences into 4 datasets: strains obtained before 2009; strains from 2009 to 2016; strains from 2017 to 2019; and strains obtained after 2019. The best fit model was selected using ModelFinder in PhyloSuite (Version 1.2.2) [52,53]. The maximum likelihood (ML) phylogenetic trees were conducted with IQTREE in PhyloSuite (Version 1.2.2), and reliability was evaluated by performing the ultrafast bootstrap replicates of 10,000 [53,54].

### 4.4. Recombination Protein Expression and Purification

The nuclear localization signal (NLS)-deleted Cap genes of PCV2a, PCV2b, PCV2d, PCV3a and PCV3b were amplified from PCV2/PCV3-positive samples, and primers are listed in Table 4, while the NLS-deleted Cap genes of PCV4 were synthesized according to the reference sequence: PCV4 HNU-AHG1-2019 (MK986820), with Pac I (New England Biolabs, Inc., Ipswich, MA, USA) and Bam HI (New England Biolabs, Inc., USA) restriction sites. The PCR reaction for PCV2 and PCV3 Cap proteins was prepared at a total volume of 25 µL: 5 µL of 5 × Q5 reaction buffer (New England Biolabs, Inc., USA), 0.5 µL of Q5 High-Fidelity DNA Polymerase (New England Biolabs, Inc., USA), 1.25 µL of each primer, 3 µL of DNA and 14 µL of ddH_2_O. The thermal conditions were initiated at pre-denaturation of 98 °C for 2 min, followed by 40 cycles of 98 °C for 3 min, 65 °C for 30 s and 72 °C for 50 s, as well as a 5 min extension at 72 °C.

Fragments of Cap genes were flanked by the Pac I (New England Biolabs, Inc., USA) and Bam HI (New England Biolabs, Inc, USA) restriction enzymes and inserted into the Pac I/Bam HI digested vector pET-32a(+) (preserved in our lab). The recombinant proteins were C-terminally fused to a Strep-tag. Then, the recombinant plasmids were transformed into *Escherichia coli* BL21 (DE3) (Vazyme Biotech Co., Ltd., China) competent cells. Subsequently, recombinant proteins were expressed under the conditions of 5 mM of IPTG (Biosharp Biotechnology, Beijing, China) at 16 °C for 24 h. The proteins were purified with Strep-tag^®^ II (IBA Lifesciences, Inc., Göttingen, Germany) and further analyzed via sodium dodecyl sulfate polyacrylamide gel electrophoresis (SDS-PAGE) and Western blot analysis.

### 4.5. Western Blot Analysis

For Western blot analysis, 10 µg of the protein was blotted onto a 0.45 µm polyvinylidene fluoride (PVDF) membrane (Absin Bioscience Inc., Shanghai, China). Membranes were incubated in 5% blocking buffer (0.05% PBST (20× PBS buffer, Sangon Biotech Co., Ltd, China; Tween-20, XILONG SCIENTIFIC, Guanzhou, China) containing 5% skimmed milk powder (Protifar, Nutricia, Zoetermeer, The Netherlands)) at 37 °C for 2 h. After blocking, membranes were incubated with primary antibodies (anti-Strep-Tag II monoclonal antibody, 1:2000, (Abbkine Scientific Co., Ltd., Atlanta, GA, USA), PCVs-positive sera, 1:100), followed by Horseradish Peroxidase (HRP)-conjugated Rabbit anti-mouse IgG or HRP-conjugated Rabbit anti-pig IgG (Sangon Biotech Co., Ltd., China; 1:1000). Finally, the membranes were examined using SuperSignal West Pico PLUS (ThermoFisher Scientific Inc., USA) according to the manufacturer’s instructions.

### 4.6. Enzyme-Linked Immunosorbent Assay (ELISA)

Next, 96-well Maxisorp microtiter ELISA plates (ThermoFisher Scientific Inc., USA) were coated with 200 ng of recombinant proteins pre-well at 4 °C for 16 h. After three washes with washing buffer (PBS containing 0.05% Tween-20, 0.05%PBST), plates were blocked at 37 °C for 2 h with 200 µL of 5% blocking buffer (0.05% PBST containing 5% skimmed milk powder (Protifar, Nutricia, Zoetermeer, The Netherlands)). After blocking, plates were washed three times with washing buffer, and the serum samples were tested in duplicate at 1:400 dilution in blocking buffer and incubated at 37 °C for 1 h. After being washed four times with washing buffer, the plates were incubated with 1:4000 diluted HRP-conjugated goat anti pig IgG H&L (Abcam, Cambridge, UK) at 37 °C for 1 h. Then, the peroxidase reaction was visualized using tetramethylbenzidine-hydrogen peroxide (TMB) solution as the substrate (GBCBIO Technologies Inc., Guangzhou, China) for 10 min. The reaction was terminated with 12.5% H_2_SO_4_, and optical densities (OD) were measured at 450 nm using a microplate reader (Bio-Rad Laboratories, Inc., USA). Both the mean average value (χ) and the standard deviations (SDs) of negative sera were used to determine the cut-off value. The OD_450_ cut-off value = χ +3SD.

## 5. Conclusions

In conclusion, we identified the predominant PCV genotypes in Southwest China, which are PCV2d and PCV3a. In the meantime, we also successfully constructed Cap proteins of different genotypes for the detection of PCV antibodies and provided comprehensive serological evidence of the infection of pigs with PCV2, PCV3 and PCV4 in Southwest China for the first time. The observation as such was in agreement with our phylogenetic analysis, while the development of serological methods based on recombinant Cap proteins also enables the PCV genotyping and immunogenicity evaluation of novel vaccine candidates.

## Figures and Tables

**Figure 1 ijms-24-15850-f001:**
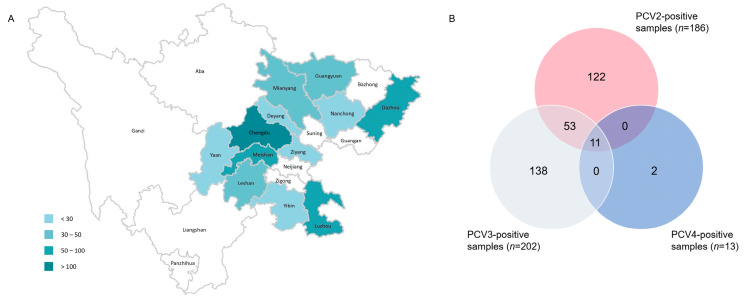
The geographical distribution of samples from Sichuan Province. The number of samples ranged from 3 to 211 (**A**). The number of positive samples detected via SYBR Green-based real-time PCR assay (**B**).

**Figure 2 ijms-24-15850-f002:**
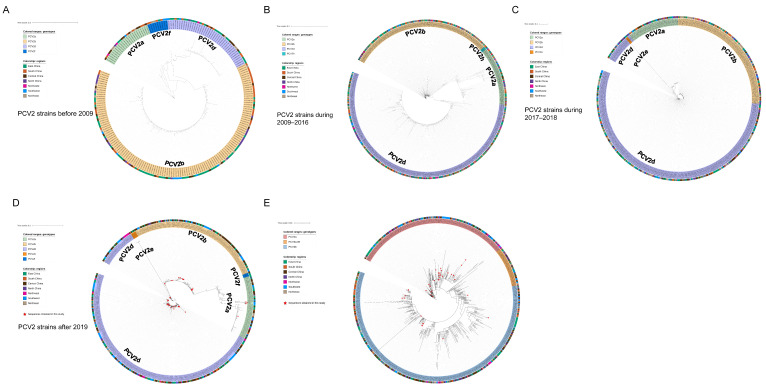
ML trees of PCV2 ORF2 gene. Scale bar: 0.1. (**A**) ML tree of ORF2 from PCV2 strains before 2009 (model: TN93 + G4). (**B**) ML tree of ORF2 from PCV2 strains during 2009–2016 (model: TIM3 + R4). (**C**) ML tree of ORF2 from PCV2 strains during 2017–2018 (model: TIM3 + R3). (**D**) ML tree of ORF2 from PCV2 strains after 2019 (model: TIM3 + R3). (**E**) ML tree of PCV3 ORF2 gene. Scale bar: 0.01 (model: TN93 + R3).

**Figure 3 ijms-24-15850-f003:**
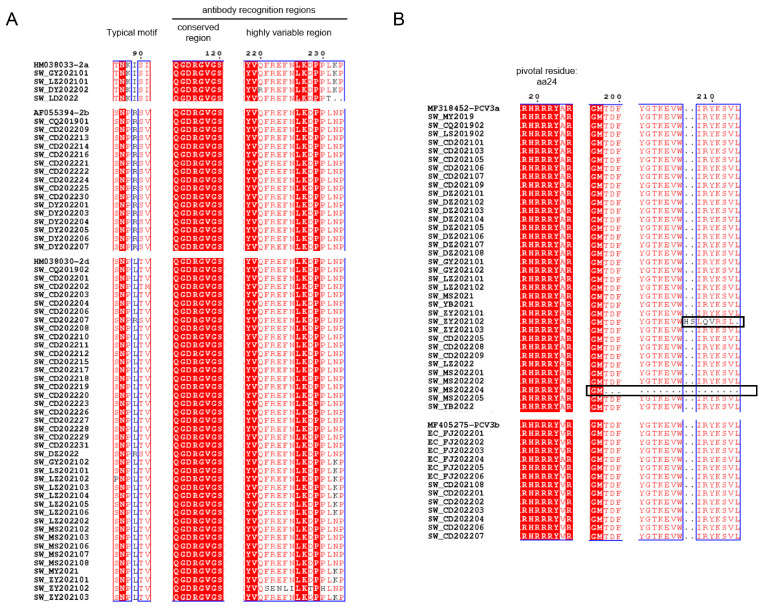
Amino acid sequence alignment of PCV2 (**A**) and PCV3 (**B**) in Cap protein. The black boxes show the residues’ deletion or frameshift mutation. The conserved amino acid sites are marked with the red background and blue boxes.

**Figure 4 ijms-24-15850-f004:**
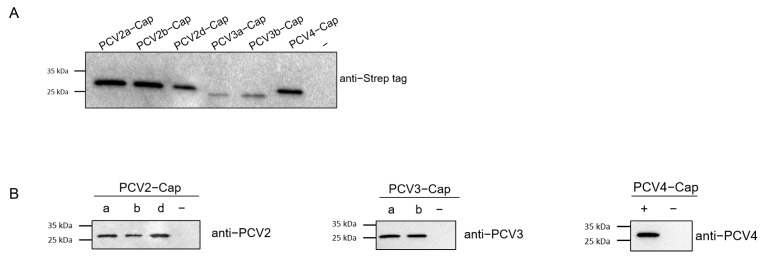
Expression and identification of PCV2, PCV3 and PCV4 Cap protein. PCV2a, 2b, 2d, PCV3a, PCV3b and PCV4 Cap proteins were identified via Western blot using anti-Strep tag antibody (**A**) and PCV2-, PCV3- and PCV4-positive sera (**B**). −: negative control, a, b, d: sub genotype of PCV2 and PCV3.

**Figure 5 ijms-24-15850-f005:**
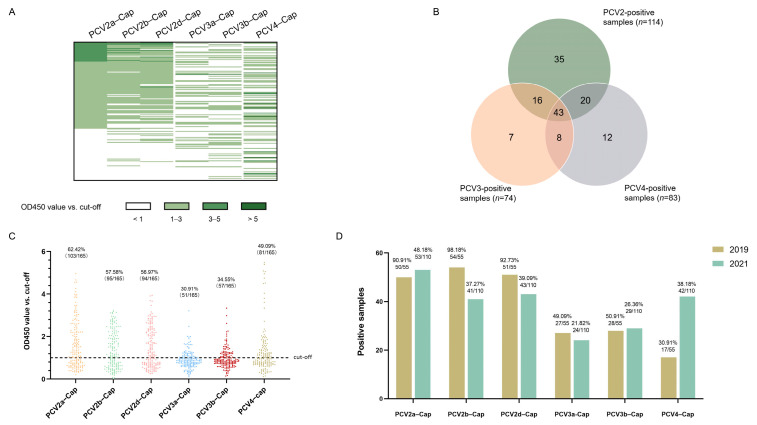
Seroprevalence of antibodies in 2019 and 2021. The OD450 values of all serum samples (*n* = 165) are displayed as a heatmap (**A**), and the total number of positive samples are shown in a Venn diagram (**B**). The distribution dot plot showed the positive rate of each antigen (**C**), while the annual positive rate is shown in a bar chart (**D**).

**Table 1 ijms-24-15850-t001:** Prevalence of PCV2, PCV3 and PCV4 of different sample types during 2019–2022.

Sample Type	PCV2 Positive Rate Positive/Total (%)	PCV3 Positive Rate Positive/Total (%)	PCV4 Positive Rate Positive/Total (%)
Serum	89/519 (17.15%)	131/519 (25.24%)	10/519 (1.93%)
Tissue	97/139 (69.78%)	71/139 (51.08%)	3/179 (2.16%)
Total	186/658 (28.27%)	202/658 (30.70%)	13/658 (1.98%)

**Table 2 ijms-24-15850-t002:** Temporal prevalence of PCV2, PCV3 and PCV4 during 2019–2022.

Year	PCV2 Positive Rate Positive/Total (%)	PCV3 Positive Rate Positive/Total (%)	PCV4 Positive Rate Positive/Total (%)
2019	9/55 (16.36%)	28/55 (50.91%)	0
2021	51/380 (13.42%)	82/380 (21.58%)	0
2022	114/223 (64.57%)	116/223 (52.02%)	13/223 (5.83%)
Total	186/658 (28.27%)	202/658 (30.70%)	13/658 (1.98%)

**Table 3 ijms-24-15850-t003:** Identities of amino acid sequences of different expressed PCV Cap proteins.

Sequence No.	Proteins	% Amino Acid Sequence Similarity				
		1	2	3	4	5
1	SW_LZ202101-PCV2a					
2	SW_CD202202-PCV2b	89.05				
3	SW_LZ202102-PCV2d	90.05	97.51			
4	SW_CD202205-PCV3a	23.47	23.47	23		
5	EC_FJ202204-PCV3b	23.94	23.94	23.47	98.91	
6	CC_MK986820_2019-PCV4	44.66	43.69	43.2	21.63	22.12

**Table 4 ijms-24-15850-t004:** Sequences of primers used in this study.

Primer	Sequences	Length (bp)
PCV2-cap-F	5′-CAGACCCCGTTGGAATGG-3′	949
PCV2-cap-R	5′-AATACTTACAGCGCACTTCTTTCG-3′	
PCV3-cap-F	5′-CTTGTGTACAATTATTGCGTTGG-3′	883
PCV3-cap-R	5′-AATACTAGCCCGGCACCA-3′	
p-PCV2a-F	5′-aaaaaGCTAGCACACCGCTACCGTTGGAGAAG-3′	603
p-PCV2a-R	5′-aaaaaTTAATTAATTAGGGTTTAAGTGGGGGG-3′	
p-PCV2b-F	5′-aaaaaGCTAGCACACCGTTACCGCTGGAGAAG-3′	
p-PCV2b-R	5′-aaaaaTTAATTAACTAAGGGTTAAGTGGGGGG-3′	
p-PCV2d-F	5′-aaaaaGCTAGCACACCGTTACCGCTGGAGAAG-3′	
p-PCV2d-R	5′-aaaaaTTAATTAATTAGGGTTTAAGTGGGGGG-3′	
p-PCV3a-F	5′-aaaaaGCTAGCAACTGCTGGAACATATTA-3′	492
p-PCV3a-R	5′-aaaaaTTAATTAATCATTCCAGTTTTTTCCGGGACAT-3′	
p-PCV3b-F	5′-aaaaaGCTAGCAACAGCTGGCACATACTA-3′	
p-PCV3b-R	5′-aaaaaTTAATTAATTAGAGAACGGACTTGTAGCGAAT-3′	

Note: Restriction endonuclease sites (underscore) protect bases (lower case).

## Data Availability

All data analyzed during this study are available from the corresponding author upon reasonable request.

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
