# Peer review of "Phylogenetic Analysis and Serological Investigation of Porcine Circovirus Indicates Frequent Infection with Various Subtypes"

_ijms, 2023, doi:10.3390/ijms242115850_

Round 1

Reviewer 1 Report

The manuscriptrticle  describes the occurrence of various subtypes of procine circovirus in China as well as its seroprevelence in the pig population.  It raises an important issue as uncontrolled outbreaks of viral diseases can cause serious consequences for pig farming. Therefore, understanding the genetics of viruses that threaten animals is very important. The work is written in a clear and acceptable language. Only some figures are illegible (figure 2, figure 5). I suggest to add some future perspective: does results encourage to develop a new vaccine? Is vaccination justified in terms of financial gains/loss?

More specific comments:

line 50 1769-1776

Introduction:

What percentage of infected animals get sick? How serious it is for swine industry?

line 82 were not constantly increasing? please change to "stable"

Methods:

Why different tissues were collected? does not it interferre with the results?

Results:

 Was the prevelence of different PCV genotypes correlated to seroprevelance?

Discussion:

line 118 healthy and disease pigs? please rewrite the sentence

Did the vaccination in 2009 improved the situation in swine industry?

line 207-209 this sentence is unclear please rewrite it

Reviewer 2 Report

The objective of this paper was to identify immunogenicity of PCVs prevalence and determination of its major circulating genotypes. Currently, well known PCV2 virus is still dangerous pathogen, but we have vaccine to protect pigs. Porcine circovirus type 3 and 4 are a newly emerging pathogen, but lack of commercially available vaccine. Serological investigations showed high PCV2 and PCV4 (50%) and PCV3 (30%-35%). The occurrence of PCV2 is well known, but PCV3 and PCV4 were described in few articles. The results obtained shall benefit the current understanding of PCV epidemiology.

Major comments:

Material and methods - total 658 samples were collected from piglet, sows or swine in some specific age? Moreover, if detection of PCV3 and PCV4 was performed by SYBR Green-based real time PCR, for PCV2 same method should be used.  Serum samples collected from Specific-pathogen free pigs should also been screened for PCV3 and PCV4, like tissue samples. Lack of DNA isolation method from serum. Lack of positive and negative controls in PCR. Lack of PCR condition in Recombination protein expression method.

Lack of name and provider  of reagents: SYBR Green reagents, primers, restriction enzymes PacI and BamHI, vector pET-32a(+), E. coli BL21(DE3), IPTG, Tween-20, PBST

Lack of name of equipment: Real-time thermocycler, electroforesis, UV Gel Documentation System, Chemiluminescent Gel Documentation

Minor comments:

Lines 30, 31, 32, 37, 39, 46, 49, 51, 53, 119, 131, 134, 144, 151, 158, 175, 194, 199, 201, 225, 227, 234, 247, 252, 253, 258 - PCV4[3], 1974[4], pigs[5], PCV3[10], PCV4[11], PCV2a-PCV2h[1], PCV4b[13,14], (PCV4)[1], replication[15,16], transmission[37-40], China[41], immunosuppression[42,43], (Figure 2A)[24], strains[45], IM group[47], routes[17], ORF2[43,52], PMWS[53], vaccines[54], ously[13,29], PCR assay[13,29], previously[13,29], analysis[30], around 2009[24], in 2017[31], of 10,000[33,34] - lack of space before [

Line 57 -  [18-21] instead of [18] [19] [20]. [21]

Line 58 – lack of reference

Line 61 - in China [17,22-25]. However…instead of in China. [17,22,23] [24,25] However…

Line 272 - Escherichia coli instead of Escherichia. Coli

Line 295 - Why serum samples were tested at 1:400 dilution?

Line 275 – if Nanodrop was used to evaluation of concentrations and purity of DNA, please add this text
